# Community-based surveillance system structures and knowledge level of community-based surveillance volunteers on priority diseases in Ghana

**Benjamin Adjei[1], Thomas Hormenu[2,3*], Joseph Kwame Mintah[2]**

**1** Regional Health Directorate, Central Region, Ghana, **2** Department of Health, Physical Education and Recreation, University of Cape Coast, Ghana, **3** Cardiometabolic Epidemiology Research Laboratory, University of Cape Coast, Ghana

\* thormenu@ucc.edu.gh

## Abstract

Community-based disease surveillance (CBS) is the backbone of the disease surveillance system in Ghana. The activities of Community-based Disease Surveillance Volunteers (CBSVs) in the Central Region have had a positive impact on the health of community members. However, little is known about the Surveillance System Structures (SSSs) and the CBSVs' knowledge levels on the various priority diseases and events. We used a cross-sectional descriptive survey to ascertain the status of SSSs and the knowledge level of CBSVs on priority diseases in the Central Region of Ghana. An ethical clearance was sought from the University of Cape Coast and Ghana Health Service. A sample of 1,381 communities were systematically sampled from 2,232 communities for this study. A structured questionnaire was used to collect data from the CBSVs within 12 weeks and was analyzed using Statistical Package for Social Sciences (SPSS). The results show that (35%) of the communities have no CBSVs, 48% have no registers, and 40% have not submitted any report for the past 1 year. Generally, (54%) of the CBSVs have low knowledge about priority diseases and events compared with the recommended national standards of 80% or more. Based on the findings of the study it could be concluded that there are weak community-based surveillance system structures and low CBSV knowledge of priority diseases and events in some communities in the Central Region of Ghana which could put the region at a higher risk of recording outbreaks that can result to morbidities, deformities and mortalities. This study recommends a multi-sectorial approach to ensure effective implementation of the Community-Based Surveillance (CBS) system. Stakeholders such as UNICEF, WHO, CDC, GHS MMDAs, and community members should collaborate to ensure improved recruitment, reporting systems, training, supervision, and provision of incentives for CBSVs to strengthen the CBS system and prevent outbreaks.

## Introduction

Community-based disease surveillance (CBS) is an approach that involves local communities in the monitoring and reporting of health events, particularly infectious diseases. This method

**Data availability statement:** Data are available within the Figshare database: 10.6084/m9.figshare.28560005

**Funding:** The authors received no specific funding for this study.

**Competing interests:** The authors have declared that no competing interests exist.

recognizes the valuable insight and immediate data that community members provide regarding health issues in their area [1]. Globally, CBS is characterized by the active participation of local populations including community health workers, volunteers, and residents. These community members are trained to collect data on health conditions and outbreaks using standard case definitions [1]. The community-based nature of the system allows for faster identification of priority diseases, better preparedness, and quick response. The CBS system effectively integrates with the health sector and the community to enhance the implementation of strategies for addressing health problems [2]

The CBS system has been used worldwide. Several African countries such as Nigeria, Kenya, Uganda, Ghana etc. have used the CBS system to combat diseases such as Guinea worm, Poliomyelitis, cholera, and malaria [2,3,4]. In Latin America countries like Brazil and Columbia used the CBS system to monitor Zika and dengue diseases [5,6]. Also, in Asia, India has successfully employed community-based approaches to monitor outbreaks of infectious diseases, employing technology and traditional methods to gather and report data [6]. The CBS system serves as an essential component of the global public health framework, especially in resource-limited settings. In Ghana, the CBS system is considered part of the entire surveillance system. It is described as the "backbone" of disease surveillance [7]. The community is the basic unit where the cases are usually detected and sometimes managed and by involving the CBSVs, community ownership for health programmes is adequately sustained. The role of the CBS system in the communities cannot be underestimated. The system helps in case identification, reporting, investigation, and confirmation. It also helps in data analyses, providing responses to outbreaks, and giving feedback about diseases or events of public health concern to the health sector and the communities in which they serve [8].

An effective CBS system requires that at the community level, structures that are efficient and sensitive enough to detect and report diseases of epidemic potentials and priority events be laid down [1]. Galimore et al. indicated weak community systems and engagement structures and hence recommended continuous assessment and strengthening of the community system structures to improve covid 19 prevention and control measures during the pandemic [9]. The Ghana Health Service (GHS) Integrated Disease Surveillance and Response (IDSR) recommends that the volunteers should be residents of the community; should be well-known; respected and accepted by the community; and should be gainfully employed. Also, they should be literate enough to record events/data on the register provided, however, an illiterate volunteer can have a literate assistant [2]. Where ethnic and religious differences exist and are likely to be a problem, the groups must be identified and involved in the selection. The selected community members are trained on key competencies to identify priority diseases and events and introduced to the community members and the opinion leaders. Furthermore, CBSVS should be presented with tools such as registers to report to the health sector [10]. Lack of these structures can result in weak surveillance which could lead to potential outbreaks. Studies have shown that community involvement and participation in the early detection and management of outbreaks is paramount to effective disease control; hence, the effectiveness of community-based disease surveillance cannot be overemphasized [11].

The priority diseases are many at the health facility level. However, in Ghana, at the community level, a few of priority disease have been selected for the CBSVs' to have adequate knowledge of them, monitor and report on within 24 hours [2] when a case is identified. They are measles, Acute Flaccid Paralysis (AFP)/Poliomyelitis, Guinea worm, neonatal tetanus, meningitis, and cholera. The events are births, infant deaths, pregnancy-related deaths, and other deaths. The rest of the events are human illness or deaths after exposure to animals, any person bitten by a dog, or cat, an unexpectedly large number of children absent from school, and any event that causes public anxiety [2] The IDSR technical guidelines indicates that

CBSVs should have an 80% or more knowledge on all the priority disease and events that they monitor and report on [2].

Knowledge level of CBSVs on Measles, AFP, and neonatal tetanus is paramount for efficient case identification [12]. According to Adokiya et al., efficient and prompt identification and management of cases could be achieved when focal persons have adequate knowledge of the situation they are dealing with [11]. A cross-sectional study done among United Arab Emirates residents above 18 years indicated that 94% of the population sampled had heard of measles but only 23% had high knowledge of the disease. Individuals with higher educational levels had more knowledge of various aspects of the disease [12]. On the part of AFP, results in the evaluation of knowledge of community leaders on AFP case detection and reporting in Nigeria posited that about 79% of the people did not know about the case definition of AFP [13]. Also, in Ethiopia, another survey indicated that Community Volunteers' knowledge of vaccine-preventable diseases was suboptimal [14]. Further on neonatal tetanus a cross-sectional study in Alexandria among females attending health offices for services, 84% of the women had very poor knowledge of maternal neonatal tetanus [15]. In another study in Nigeria, conducted among young women of childbearing age, the majority of respondents 64%, had poor knowledge of the causes and risk factors for Neonatal tetanus [15].

CBSVs knowledge of Guinea worm, cholera, and meningitis varied across countries. A cross-sectional study conducted in Juba in the Central Equatorial State indicated that about 56% of respondents know the symptoms and the cause of Guinea worm [16]. Also, a cross-sectional study conducted in the Osiolo community in Kenya indicated that only 53% exhibited a higher knowledge of the cholera disease. In Ghana, an assessment of response to the cholera outbreak in two districts namely Akatsi district in the Volta Region and Komenda Edina Eguafo Abriem (KEEA) district in the Central Region revealed that inadequate community knowledge about the disease and its preventive measure are some of the contributing factors of its spread [17]. In another study within the Ghanaian meningitis belt, it was realized that about 50% of the respondents understood the role of whether in the cause of meningitis which has implications on transmission, prevention, and management of the disease [2]. Wuilfan et al, also indicated inadequate knowledge of CSM diagnosis in cross-border towns in the Upper East region in Ghana [18].

The World Health Organisation (WHO) posits that indicators for the evaluation of surveillance systems should be identified from a broader component of surveillance, including the surveillance system structures, core functions, support functions, quality functions, and priority diseases [19]. The surveillance system structures look at how well the surveillance activities comply with the standards, regulations, and laws, key strategies, networking and partnership and involvement of stakeholders. The structures ensure that the operational procedures are adhered to improve the quality of the surveillance system. The core functions measure the system's process and output, including case detection, case registration, case confirmation, and reporting. The rest are data analysis and interpretation, epidemic preparedness, response and control and feedback [2] The interplay of these functions provides an effective surveillance system. This study focuses on the surveillance system structures and priority diseases and events.

Preliminary records reviewed by the research team at Abura Aseibu Kwamansese District in the Central Region indicated that about 65% of the communities were not under active surveillance and among some of the potential factors mentioned were lack of supervision and resources to enhance the CBSV's work. However, these factors have not been studied and documented. In addition, monitoring reports show inadequate knowledge of CBSVs on priority diseases and events. There is a non-availability of CBSVs in some of the communities which have also not been investigated by earlier studies [9]. Moreover, differences in the

knowledge level of CBSVs on priority diseases and events and supportive functions were not the focus of previous studies [20]. The purpose of this study was to evaluate the CBS system structures in the communities, assess the knowledge level of the CBSVs on case definitions of priority diseases and health events in their registers, and determine the differences between the knowledge level of CBSVs and supportive functions in the Central Region of Ghana. The study therefore sought to determine the status of the surveillance system structures in the Central Region of Ghana, Ascertain the knowledge level of CBSVs on the priority diseases and events in their registers, and determine whether the association between the knowledge level of CBSVs on priority diseases and supportive functions.

## Methods

### Study design

A cross-sectional descriptive study design was used. This design was employed because the data were collected at one point in time to answer the research questions [2,82,8]. The cross-sectional study has varied strengths. Fraenkel et al. [21] posit that, in cross-sectional surveys, a wide range of data are collected on people's characteristics to make sound generalizations. It helps to make health-policy decisions. Hence, this design was appropriate because data were collected from a large number of CBSVs. However, a key weakness of the cross-sectional study was that it could not distinguish between long-standing and newly occurring situations. The data were collected within 12 weeks to determine the status of the surveillance system structures in the communities and the knowledge level of CBSVs on priority diseases and events. It was also used to ascertain the differences between the knowledge level of CBSVs and on core and supportive functions of CBSVs. Fraenkel et. al posits that, in cross-sectional surveys, a wide range of data is collected on people's characteristics [21]. Hence, this design was appropriate because data was collected from many CBSVs.

### Study area

The study was carried out in the Central Region of Ghana. The region shares a border on the East with the Greater Accra Region, on the North with the Ashanti Region on the North-East with the Eastern Region, and on the West with the Western Region. The region has 22 administrative districts, 112 sub-districts, 343 functional Community-based Health Planning and Services (CHPS) compounds and 2,232 communities [22]. These levels are well-organized facilities of districts, sub-districts, and CHPS level and provide a very good opportunity for good system structures and provision of adequate knowledge of CBSVs on priority diseases [7]. From the 2020 population census, an estimated population of 2,989,594 was projected for the year 2023, with an annual growth rate of 2.4%. The population of children between 0-11 months constitutes 4% of the total population which stands at 119,598. The region was selected because it has potential to report outbreaks. In 2016, there was an outbreak of cholera in the region of which over 100 cases were recorded [23]. In 2023 the annual report of the Central Regional Directorate indicated that there were 6 confirmed meningitis cases and 15 confirmed measles cases. There were also 4 suspected yellow fever cases, 1 monkeypox case, 88 confirmed Covid-19 cases, and 43 New flu illnesses. The above situation makes the region a relevant area for the study [23].

### Sampling strategy and sample size

A total sample size of 1,381 CBSVs comprising males and females was used for the study. The sample size for the CBSVs was estimated based on the following references: Ogah [24] posited that, with a degree of accuracy = +/- 0.05; a proportion of sample size 0.5; and a

confidence level of 95%, a sample of 328 could be drawn from a population of 2, 200 which is 15% of communities in the Central Region. Hamalaw et al. [12] in evaluating communicable disease surveillance in Iraq, used 25% of the facility's focal persons for the study. Aziz et al. [25] in their study recommended that in evaluating a programme, a statistically realistic and well-representative figure must be considered. Afari-Asiedu et al. [26] in a community study, sampled 50% and more of the total communities, and a focal person was selected from each of the communities. However, for comprehensive research sense, in this study, a total of 1, 381 communities were sampled from 2,232 communities in the region, and the CBSVs residing in these communities were used for this study, representing about 62% of the communities in the region to increase the power of the study.

The sampling processes started from the district level, where, in each of the districts, with the support of the DHMT all the communities were listed, and a systematic sampling (selection from the list at fixed intervals) was used to select 62% of the communities [26]. In each of the selected communities, the research assistant visited the community, and the Community Health Nurse (CHN) in charge was contacted for details of the community in terms of whether there was a CBSV or not. Where there were no CBSVs, the CHN was asked about the name of the community, the developmental and ecological status, and the reasons for the non-availability of a CBSV in the community. Where there were CBSVs, they were selected to participate in the study.

## Data collection techniques

A structured questionnaires was used to collect data from the CBSVs. WHO's framework and the generic questionnaire for monitoring and evaluating surveillance and response systems for communicable diseases were adopted and used [19]. The framework was developed by experts of WHO and has also been used to evaluate the national surveillance system in Sudan and the Kurdistan Region in Iraq [8,12,19,20].

To ensure the validity of the instruments, the researcher ensured that the items on the questionnaire represented the domain of interest. Again, the question items were reviewed by colleagues and experts in the Disease Surveillance Unit and academia for scrutiny, corrections and clarity for face and content validity. In the questionnaire, some sections focused on broader areas of the research objective to elicit information from the CBSVs. The questionnaire had Sections A to D. Section A focused on the background information and socio-demographic data of the CBSVs. Section B examined the availability of surveillance systems structures that have been put in place in the community. Section C posed questions to assess the CBSVs' knowledge on priority diseases and vital health events. The CBSVs were asked to state the standard case definitions of priority diseases and events in their register and the responses were scored and analysed on a scale of 0 to 3, and Section D documented whether the CBSVs have received support from the higher level in terms of training, resources, supervision, communication, and coordination.

A pretest of the instrument was conducted to minimise errors of the instruments used for the data collection. The questionnaires were pre-tested in two districts; at Shama and Daboase districts (coastal and forest districts respectively) in the Western Region of Ghana, which has similar characteristics as most districts in the Central Region. The instrument was administered to 40 CBSVs. Experiences from the pretest were used to revise the wording and arrangement of the questions. Cronbach's Alpha was used to calculate the internal consistency of the items in the questionnaires at the pretest and real research stages which gave a reliability coefficient as $\alpha = 0.89$ and $\alpha = 0.98$ respectively. These agreed with Fraenkel et al. [21] who opined that a reliability co-efficient of 0.7 or above is enough for a study, indicating that the instrument for the study was highly reliable.

At the community level, the research assistants went to the chief's palace or the Community Health Nurse (CHN) in charge of the CHPS compound and made inquiries about the presence of CBSVs in the community. If there was a volunteer, the volunteer's consent was sought and upon agreement, the questionnaire was administered by the research assistant to the CBSVs. In the absence of CBSVs, a portion of the questionnaire was filled with the help of the Community Health Officer (CHO) and the reasons for absence of CBSVs were captured. Data was collected during the day between 7:00 a.m. and 6:00 pm and within 3 months from April 15, 2020 to July 15, 2020.

## Training of data collectors

In each of the districts, the researcher was introduced to the DHMT, and with their support two national service personnel residing in the district were recruited and trained as research assistants to support the data collection. The training content included an overview of the CBS system in the Central Region, rationale for the study, a case definition of the priority diseases and events in the CBSVs register, a discussion of the CBSVs register, a discussion on the questionnaire items, a practical session on questionnaire administration and a fieldwork. The training sessions were carried out within four (4) hours. Lunch was provided to the trainees. This was to help research assistants to understand the content and do a proper assessment of responses from participants to reduce biases of all forms. The training was decentralized to help reduce the cost of accommodation and other expenses. It was also to ensure that travel time and revisits of participants who were not available for the first visits were considered and addressed. With the support of the DHMT, a movement plan was drawn up to guide the movements of the research assistants to the communities where the data collection was done.

## Data processing and analysis

After the data collection, the data were edited and coded and kept confidential and under a pass-worded computer and no other persons had access to them. Statistical Package for Social Sciences (SPSS) Version 21, STATA Version 17, and Microsoft Excel were used to process the data since the data was numerical [12]. The data collected were screened to determine the accuracy of the data, dealt with missing data, and checked the effects of some of the values on the analysis. Frequencies were used for each of the questions to check for errors and corrections were made accordingly. The analysis was done based on the research objectives for the study which were as follows: to evaluate the CBS system structures in the communities, to assess the knowledge level of the CBSVs on case definitions of priority diseases and health events, and to determine the differences between knowledge level of CBSVs on their core and supportive surveillance functions.

The data on the surveillance system structures were screened, and descriptive statistics (percentages and proportions) were used to describe and present the data, using frequency tables. To assess the CBSVs' knowledge level of the priority diseases and events, CBSVs were asked to give the case definition of each of the priority diseases and events in their registers. The responses were scored from *0* to *3*. If all the signs and symptoms of the specific disease/events were mentioned, *3* was scored and that was *excellent*. If more than half of the signs and symptoms of the disease/event were mentioned, *2* was scored and that was *good.* If half of the signs and symptoms of the disease/event were mentioned *1* was scored and that was an *average.* Finally, if less than half or none of the signs and symptoms were mentioned *0* was scored and that was *poor.* The scores were added up converted to percentages and presented in frequency tables. The results were interpreted on a four-level scale: Poor, Average, Good, and Excellent.

The interpretation of the scores was explained in detail as follows: *Poor* – were those who scored below 50%, which meant that CBSVs within this group *always* needed help to identify a priority disease or an event. *Average* – were those who scored from 50-69, meaning CBSVs *sometimes* need help to identify a priority disease or an event. *Good* – were CBSVs who scored within 70-79, these CBSVs *rarely* need help to identify a priority disease or event. Finally, *excellent* were those who scored from 80-100, these CBSVs *never* need help to identify a priority disease or event. The interpretations *always* and *sometimes,* and *rare* and *never* have been combined to make the discussion more meaningful [21]. See Table 1 below which summarises the scoring system.

A chi-square test was performed to determine the association between the knowledge level of participants and on core and supportive functions of CBSVs after the data were screened. Based on the assumption that the data was large, randomly selected, categorical, and have mutually exclusive categories [21]. Further logistics regression was performed on the variables that showed significant differences to determine the association. The dependent variable was further dichotomised into weak (average and poor) knowledge and better (excellent and good) knowledge and analysed This process produces odds ratio, p-value, and confidence intervals for the variable measured and was discussed appropriately.

## Ethics statement

Before the start of the data collection, an introductory letter was provided by the Department of Health, Physical Education and Recreation to introduce the researcher to Ghana Health Service and Institutional Review Boards. Ethical clearance was sought from the University of Cape Coast's Institutional Review Board in Cape Coast (ID: UCCIRB/CES/2019/45), and the Ghana Health Service Ethics Review Committee in Accra (ID: GHS-ERC001/02/20) respectively. Approval was also sought from the Central Regional Health Directorate. All participants provided written informed consent after the purpose of the study had been explained to them and their right to interrupt the interview at any time or decline the study without any fear of prejudice. The names of respondents were not associated with responses provided to ensure their anonymity. Participants were informed about their freedom to skip some of the questions and exit from the study. All information obtained from the participants was kept confidential and under locked file cabinets at the offices of the investigator. There were no risks associated with the study and there were no material or financial benefits to respondents. However, participants were informed of how beneficial the information they provided, will help the region and policymakers to make meaningful decisions to improve the health system.

## Results

### Status of the community-based disease surveillance system structures in the Central Region of Ghana

**Socio-demographic characteristics of CBSVs.**  The background of participants and demographic characteristics of a study is key to the discussion of the research. The

**Table 1.  Scoring and Interpretation of CBSVs on Knowledge of Priority Diseases and Events.**

| Scores | Interpretation | Marks % | Description |
|--------|----------------|---------|-------------|
| 3 | Excellent | 80–100 | Never need help |
| 2 | Good | 70–79 | Rarely need help |
| 1 | Average | 50–69 | Sometimes need help |
| 0 | Poor | Below 50 | Always need help |

characteristics of the communities and demographic information on the CBSVs such as developmental status, ecological status, sex, age, educational background, occupation, and religion have been described in this research as follows. With regards to Table 1, the study revealed that about 85% (n = 1181) of the communities surveyed were in the rural areas while 15% (n = 200) were in the urban communities. Also, 79% (n =1094) were in the forest areas while 21% (n = 287) were in the coastal communities. The majority of the CBSVs in the communities were males 68% (n = 612) and females 32% (n = 286). The average age of CBSVs was 44.5 and the modal age group was 30-39 years. However, the oldest age was 82 years and the youngest was 20 years.

The educational background, occupation, religion, and marital status of respondents were further displayed in Table 1 as follows: participants with degrees were 1% (n = 10), Diploma 5% (n= 47), and Senior High School 29% (n = 261). The rest were Junior High School 52% (n = 471), Primary 9% (n = 76), and None 4% (n = 33). The predominant occupation of the CBSVs was Farming 57% (n = 512) followed by trading 16% (n = 144), teachers 6% (n = 53) health workers 3% (n = 27), Unemployed 9% (n=81) and others including artisans, retirees and students 9% (n = 81). Concerning the religion of respondents, Christians were 90% (n = 808), Moslems 9% (n = 81) and Traditionalists 1% (n = 9). Marital status of respondents revealed as follows: single 21% (n=190), married 71% (n=639) separated 2% (n=14), divorced 2% (n=18) and widowed 4% (n=37) (Table 2).

## Disease surveillance system structures in the communities

The community-based surveillance system has been designed to complement the efforts of the facility-based and other surveillance structures in the health sector [27]. Within the Ghana Health Service Integrated Disease Surveillance Technical Guidelines, it is recommended that every community must have a Community-Based Disease Surveillance Volunteer to liaise between the community and the health sector [2]. A surveillance system in a community is said to be well structured and functioning when all the following have been put in place: (a) the presence of at least one selected volunteer who resides in the community, (b) the volunteer must be trained on disease surveillance before he/she starts work, and (c) must be introduced to the community members. Also, the volunteer must have a CBSV register to use to compile reports and submit them to the nearest health facility every month. When all these structures are in place, then the CBS system will be sensitive enough to pick any priority condition and events as early as possible for immediate response [2].

This study tried to examine if the surveillance system structures were well established in the communities and Table 3 below presents the results. Out of the 1381 communities surveyed 65% (n=898) of them had CBSVs while 35% (n=483) had no CBSVs. For those that do not have CBSVs, the major reason why there were no CBSVs was that the CBSVs had stopped working 38% (n=184). Another key reason was that no CBSV had been selected before 31% (n=150). Other key reasons were as follows: CBSVs had traveled 13% (n=62), the community did not know the CBSVs system 6% (n=30), CBSVs died 4% (n=20), CBSVs gone to school 2% (n=12) and CBSVs had gone on transfer 1% (n=3). Another specified reason was that CBSVs were sometimes invited to work only when there were campaigns 5% (n=22).

Furthermore, on the surveillance system structure, the study revealed that for communities that had CBSVs, CBSVs had stayed in the communities on average for 20 years or more and they were mostly selected by their community members 61% (n=547), 30% (n=269) were selected by health workers and 7% (n=64) were also selected by self, while 2% (n=18) were selected by NGO. Generally, there was training after selection, the study indicated that 78% (n=700) of the CBSVs were trained before they started the CBSV work in their community, however, 22% (n=198) of them were not trained but they started work. After training CBSVs must be introduced to

**Table 2. Socio-demographic Characteristics of CBSVs.**

| Demographics | Frequency | Percentage |
|---|---|---|
| **Developmental Status** | | (n = 1381) |
| Urban | 200 | 15 |
| Rural | 1181 | 85 |
| **Ecological Status** | | (n = 1381) |
| Forest | 1094 | 79 |
| Coastal | 287 | 21 |
| **Sex** | | (n = 898) |
| Males | 612 | 68 |
| Females | 286 | 32 |
| **Age** | | (n = 898) |
| 20-29 | 118 | 13 |
| 30-39 | 238 | 26 |
| 40-49 | 213 | 24 |
| 50-59 | 185 | 21 |
| 60 and above | 144 | 16 |
| **Educational level** | | (n = 898) |
| Degree | 10 | 1 |
| Diploma | 47 | 5 |
| SHS | 261 | 29 |
| MSLC/JHS | 471 | 52 |
| Primary | 76 | 9 |
| None | 33 | 4 |
| **Religion** | | (n = 898) |
| Christianity | 808 | 90 |
| Moslem | 81 | 9 |
| Traditionalist | 9 | 1 |
| **Occupation** | | (n = 898) |
| Farmer | 516 | 58 |
| Trading | 140 | 16 |
| Teacher | 49 | 5 |
| Health Worker | 29 | 3 |
| Unemployed | 81 | 9 |
| Artisans | 83 | 9 |
| **Marital status** | | (n = 898) |
| Single | 190 | 21 |
| Married | 639 | 71 |
| Separated | 14 | 2 |
| Divorced | 18 | 2 |
| Widowed | 37 | 4 |

Source: Field Data (2023)

the community members. The majority, 86% (n=772) of CBSVs had been introduced into their communities but about 14% (n=126) were not introduced to the communities before they started working. On availability of registers, 62% (n=557) of CBSVs had registers but 38% (n=341) had no registers. In addition, 60% (n=539) of the CBSVs have not submitted any report to the next level for the past year and only 40% (n=359) had submitted some reports.

**Table 3. Disease Surveillance System Structures in the Communities.**

| System structures | Frequency | Percentage (n=1381) |
|---|---|---|
| **Availability of Volunteer** | | |
| Yes | 898 | 65 |
| No | 483 | 35 |
| **Reasons for non-availability** | | (n=483) |
| Do not Know about CBSV | 30 | 6 |
| No one selected before | 150 | 31 |
| Ad-hoc Called only for NIDs | 22 | 5 |
| Volunteer died | 20 | 4 |
| Volunteer on Transfer | 3 | 1 |
| Volunteer gone to school | 12 | 2 |
| Volunteer stopped work | 184 | 38 |
| Volunteer travelled | 62 | 13 |
| **Length of Stay in Community** | | (n=898) |
| 0-5 years | 62 | 7 |
| 6-10 years | 90 | 10 |
| 11-15 years | 81 | 9 |
| 16-20 years | 135 | 15 |
| 21 and above | 530 | 59 |
| **Criteria for Selection** | | (n=898) |
| Community | 547 | 61 |
| Health Worker | 269 | 30 |
| Self | 64 | 7 |
| NGO | 18 | 2 |
| **Trained before work** | | (n=898) |
| Yes | 700 | 78 |
| No | 198 | 22 |
| **Introduced to community** | | (n=898) |
| Yes | 772 | 86 |
| No | 126 | 14 |
| **Provided Register** | | (n=898) |
| Yes | 557 | 62 |
| No | 341 | 38 |
| **Submit Report** | | (n=898) |
| Yes | 539 | 60 |
| No | 359 | 40 |

Source: Field Data (2023).

## Knowledge level of CBSVs on the case definitions of priority diseases and vital health events in the community volunteer registers in the central region of Ghana

The results on knowledge level of CBSVs on priority diseases and events are presented in Table 4. Generally, results in Table 4 indicate that about 28% (n=251) of the CBSVs had excellent knowledge of the case definition of priority diseases and events while 18% (n=162) had good knowledge. Table 4 also shows that 29% (n=260) and 25% (n=225) of the CBSVs had an average and poor knowledge of the case definition of the priority diseases and events respectively.

**Table 4. General Knowledge of CBSVs on Case Definitions of Priority Diseases and Events.**

| Scores | Interpretation | Frequency | Percentage |
|--------|---------------|-----------|------------|
| 80-100 | Excellent | 251 | 28 |
| 70 – 79 | Good | 162 | 18 |
| 50- 69 | Average | 260 | 29 |
| Below 50 | Poor | 225 | 25 |
| Total | | 898 | 100 |

Source: Field Data (2023).

Results of the CBSVs' knowledge of the specific priority diseases and events are presented in Table 5. The first in Table 5 is the reportable events. The results show that 27% (n=242) of the CBSVs had excellent knowledge of the reportable events, 40% (n=359) had good knowledge, 23% (n=207) had average knowledge and 10% (n=90) had poor knowledge of the reportable events. Second, in Table 5 is Cerebrospinal Meningitis (CSM), the result indicates that 24% (n=215) of the CBSVs had excellent knowledge of CSM, 26% (n=233) had good knowledge and 27% (n=243) had average knowledge, while 23% (n=207) had poor knowledge.

Cholera is the third priority condition. The results point out that about 56% (n=503) of the CBSVs had excellent knowledge of cholera, 28% (n=251) had good knowledge while 11% (n=99) and 5% (n=45) of the CBSVs had average and poor knowledge of cholera respectively. The fourth priority condition is neonatal tetanus, and the data shows that about 22% (n=198) of the CBSVs had excellent knowledge of the case definition of neonatal tetanus and 30% (n=269) had good knowledge. However, 29% (n=260) had an average knowledge, while 19% (n=171) recorded poor knowledge of the disease.

Results on the level of knowledge of CBSVs on measles which is the fifth priority condition in Table 5 show that about 46% (n=413) had excellent knowledge, 37% (n=332) had good knowledge, 13% (n=117) had average knowledge, and 4% (n=36) of the CBSVs had poor knowledge on measles. Furthermore, Guinea worm is the sixth condition. The data indicate 39% (n=350) had excellent knowledge, 34% (n=305) had good knowledge, while 16% (n=144) and 11% (n=99) had average and poor knowledge respectively. Finally, the seventh and last condition which is Acute Flaccid Paralysis (AFP)/Poliomyelitis, the results show that 38% (n=341) had excellent knowledge, 37% (n=332) had good knowledge, 16% (n=144) had average knowledge while 9% (n=81) of CBSVs had poor knowledge on AFP.

## Differences between the knowledge level of CBSVs and on core and supportive surveillance functions of CBSVs

Further detailed analysis was done to establish the association between the supportive surveillance functions and the dependent variable (knowledge). As shown in Table 6, respondents' knowledge of priority diseases and events established an association with training on surveillance (p<0.001), receiving resources from a higher level (p<0.001), receiving supervision from a higher level (p=0.010) and availability of surveillance committee at the community (p<0.001). However, there was no relationship between respondents' knowledge about surveillance and communication (p=0.523).

Bivariate and multivariate logistic regression of independent variables that had an association on the level of knowledge about surveillance was conducted. Table 7 presents the results of the logistic regression conducted for the association between the level of participants' knowledge of surveillance and the independent variables of the study. For the logistic regression, the dependent variable was further dichotomised into weak knowledge (average and

**Table 5. CBSVs Knowledge Level on Specific Priority Diseases and Events.**

| Disease | Frequency | Percentage |
|---|---|---|
| **Reportable Events** | | **(n = 898)** |
| Excellent | 242 | 27 |
| Good | 359 | 40 |
| Average | 207 | 23 |
| Poor | 90 | 10 |
| **CSM** | | **(n = 898)** |
| Excellent | 215 | 24 |
| Good | 233 | 26 |
| Average | 243 | 27 |
| Poor | 207 | 23 |
| **Cholera** | | **(n = 898)** |
| Excellent | 503 | 56 |
| Good | 251 | 28 |
| Average | 99 | 11 |
| Poor | 45 | 5 |
| **Neonatal Tetanus** | | **(n = 898)** |
| Excellent | 198 | 22 |
| Good | 269 | 30 |
| Average | 260 | 29 |
| Poor | 171 | 19 |
| **Measles** | | **(n = 898)** |
| Excellent | 413 | 46 |
| Good | 332 | 37 |
| Average | 117 | 13 |
| Poor | 36 | 4 |
| **Guinea Worm** | | **(n = 898)** |
| Excellent | 350 | 39 |
| Good | 305 | 34 |
| Average | 144 | 16 |
| Poor | 99 | 11 |
| **AFP/Poliomyelitis** | | **(n = 898)** |
| Excellent | 341 | 38 |
| Good | 332 | 37 |
| Average | 144 | 16 |
| Poor | 81 | 9 |

**Note: Interpretation in %: Excellent-80-100; Good- 70-79; Average- 50-69; Poor- Below 50.**

Source: Field Data (2023).

poor) and better knowledge (good and excellent). It showed that, at the bivariate level, there was a significant association (p<0.05) between knowledge of priority disease and events with receiving training and resources from the higher level and having a functional surveillance committee (coordination) at the community level. However, after controlling for the effect of other variables in the multivariate logistic regression model, receiving resources from the higher level and availability of the surveillance committee (coordination) showed a significant association with the level of knowledge on priority diseases and events. Respondents who received some form of resources from the higher levels (AOR 2.93, 95% CI 1.976-4.361,

**Table 6. Association between the supportive surveillance functions and the dependent variable (Knowledge).**

| Variables/Categories | Level of surveillance knowledge [n=898 (%)] | | | | P-value[a] |
|---|---|---|---|---|---|
| | Poor | Average | Good | Excellent | |
| **Received any training on disease surveillance** | | | | | |
| No | 127 (57.5%) | 134 (50.8%) | 76 (47.5%) | 94 (37.2%) | <0.001[*] |
| Yes | 94 (42.6%) | 130 (49.3%) | 84 (52.5%) | 159 (62.9%) | |
| **Received resource(s) from the higher level** | | | | | |
| No | 180 (81.5%) | 155 (58.8%) | 95 (59.4%) | 136 (53.8%) | <0.001[*] |
| Yes | 41 (18.6%) | 109 (41.3%) | 65 (40.7%) | 117 (46.3%) | |
| **Received supervision from the higher level** | | | | | |
| No | 128 (58%) | 144 (54.6%) | 92 (57.5%) | 112 (44.3%) | 0.010[*] |
| Yes | 93 (42.1%) | 120 (45.5%) | 68 (42.5%) | 141 (55.8%) | |
| **Received communication from the higher levels** | | | | | |
| No | 72 (32.6%) | 85 (32.2%) | 52 (32.5%) | 69 (27.3%) | 0.523 |
| Yes | 149 (67.5%) | 179 (67.9%) | 108 (67.5%) | 184 (72.8%) | |
| **Availability of a committee in the community that meets to discuss surveillance issues** | | | | | |
| No | 155 (70.2%) | 138 (52.3%) | 83 (51.9%) | 127 (50.2%) | <0.001[*] |
| Yes | 66 (29.9%) | 126 (47.8%) | 77 (48.2%) | 126 (49.9%) | |

P-value: [*]p < 0.05; [a] Chi-squared test (chi$^2$)/Fisher's exact test.

Source: Field Data (2023).

**Table 7. Odds ratios (Crude and Adjusted) from binary logistic regression for the association between independent variables that influenced level of knowledge (multivariate logistic regression analysis results).**

| Variables/Categories | Level of surveillance knowledge by respondents | | | |
|---|---|---|---|---|
| | COR (95% CI) | P-value | AOR (95% CI) | P-value |
| **Received any training on disease surveillance** | | | | |
| No | Ref | | | |
| Yes | 1.658 (1.220 - 2.253) | 0.001[*] | 1.163 (0.813 - 1.664) | 0.408 |
| **Received resource(s) from the higher level** | | | | |
| No | Ref | | Ref | |
| Yes | 3.310 (2.282 - 4.799) | <0.001[*] | 2.935 (1.976 - 4.361) | <0.001[*] |
| **Received supervision from the higher level** | | | | |
| No | Ref | | Ref | |
| Yes | 1.301 (0.958 - 1.768) | 0.092 | 0.802 (0.560 - 1.149) | 0.230 |
| **Availability of committee in the community that meets to discuss surveillance issues** | | | | |
| No | Ref | | Ref | |
| Yes | 2.220 (1.604 - 3.073) | <0.001[*] | 1.857 (1.313 - 2.624) | <0.001[*] |

P-value: [*]p<0.05; COR: Crude Odds Ratio; AOR: Adjusted Odds Ratio; 95% CI: 95% Confidence Interval; ref: Reference group.

Source: Field Data (2023).

p<0.001) and those who had committees that discuss surveillance issues (coordination) within their communities had higher odds of having better knowledge about priority surveillance (AOR 1.85, 95% CI 1.313 - 2.624, p<0.001).

## Discussion

WHO's framework for evaluating surveillance system recommends for a strong system structure at the community level where a trained CBSV will be resourced to work and report to the health authorities [28]. However, this study revealed a weak surveillance system structure in the communities in the Central Region of Ghana, as a significant number of the communities have no CBSVs. A situation that can hamper the progress of achieving the universal health coverage by 2030 [19]. This finding is consistent with a study by Galimore *et al.* [9] that indicated weak community systems and engagement structures and hence recommended for continuous assessment and strengthening of the community system structures to improve covid 19 prevention and control measures during the pandemic.

Weak community-based surveillance system structures do not only result in the delay in detection and response to potential threats on priority diseases which cause mortalities but also result in dissemination of limited health information to the community which could lead to misinformation and a lack of preparedness for health emergencies [20]. CBSVs are to liaise with the community and the health sector to facilitate organisations and implementation of health programmes and without their presence there may be a disconnect between the community and health services which could impact negatively on the overall health outcomes [2]. Weak structures such as non-availability of CBSVs, poor reporting system of CBSVs, inadequate resources provision and poor collaboration with the community affect health monitoring and response in the community [8].

The Integrated Disease Surveillance and Response Technical Guidelines for Ghana endorses CBSVs should be trained to have 80% or more knowledge on priority diseases and events for effective disease surveillance [29]. Nevertheless, this study revealed that CBSVs have low knowledge about priority diseases and events, thus, a significant number of CBSVs need help before they can identify a priority disease, especially Neonatal tetanus, and Cerebrospinal meningitis (CSM) in the Central Region of Ghana. This situation needs urgent attention because it can contribute to high neonatal and infant mortality rates in the region [15]. This finding agrees with the study amongst young women in Nigeria on preventive practices against neonatal tetanus which indicated that a significant number of respondents have poor knowledge of neonatal tetanus [30].

Consequently, low CBSVs knowledge on priority diseases and events could contribute to the delays in case detection and reporting in the communities which could result in epidemics [30]. Also, inadequate knowledge of priority diseases could lead to ineffective disease surveillance and reduced community engagements, increased spread of misinformation and higher rates of morbidities and mortalities on priority diseases [19]. Some of the reasons for the low knowledge have been documented as a lack of consistent training due to inadequate funds and poor staff knowledge on the community-based surveillance programme [31]. More resources should be channeled to build the capacity of the health staff and CBSVs on the CBS programme and priority diseases and events.

The study also indicated that CBSVs who received some form of resources from the higher levels and those who had committees that met and discussed surveillance issues within their communities had higher odds of having better knowledge about priority diseases and events. This is in line with a study in Chad which revealed that communities visited by volunteers at least twice weekly had better knowledge of Guinea worm symptoms and could name more prevention strategies than villagers visited less frequently [30,31]. Those who received resources and had meetings had a better knowledge of priority diseases and events than those who did not. This could be because periodically during meetings priority diseases are discussed and this serves as an orientation to the CBSVs. The implications of a lack of resources and review meetings led to a weak understanding

of CBSVs roles, decreased motivation, burnout, and a decline in knowledge retention and application [32,33].

The study has established that weak surveillance system structures and low CBSVs knowledge on priority diseases and events are affecting the community-based surveillance system in the Central Region of Ghana. Interestingly, the study has further revealed that CBSVs who were better resourced and had the opportunities of meeting and discussing issues on surveillance had higher chances of better knowledge on priority diseases and events than their counterparts. The consequences of these findings could have a detrimental effect such as potential epidemics on the region and the nation at large. More significantly, these findings could guide the Ghana Health Service, health partners and other relevant stakeholders to design and formulate policies to strengthen the CBS system in Ghana.

## Conclusion

There are weak community-based surveillance system structures and low CBSVs knowledge of priority diseases and events in the Central Region of Ghana. This situation put the region at a higher risk of recording outbreaks which could lead to morbidities and mortalities. Measures such as stakeholder collaboration, capacity building of staff and CBSVs on CBS system and priority diseases, continuous quarterly reviews, and timely provision of appropriate resources to CBSVs could help improve the implementation of the CBS programme.

## Recommendations

Stakeholders such as Ghana Health Service and the Ministry of Health and health partners such as UNICEF, CDC and WHO and opinion leaders must continuously be brought on board to periodically review and evaluate the CBS system structures and develop contemporary ways to implement the CBS system. The DHMT should build the capacity of the health staff on the guidelines on the implementation of CBS activities so that they can also continuously train the CBSVs on the priority diseases and events to perform their roles effectively and efficiently. DHMTs should establish a functional coordination committee at the district level to be responsible for all CBSV activities including recruitment and reporting. This committee will be responsible for planning, budgeting, implementing, and evaluating the core, supportive, and quality surveillance functions and instituting innovative ways to motivate the CBSVs. Ghana Health Service and the Ministry of Health should develop a policy to regulate the activities of the CBSVs.

## Limitations

The inaccessible nature of some of the roads to the communities made it impossible to use four-wheel drive and, in this case, motorbikes and bicycles were used as an alternative to the communities. The sample size was very large and arrangements for meeting times and seeking consent from all the respondents made the process more laborious, however, the use of easy-to-understand questionnaires helped in addressing these challenges.

## Acknowledgments

We acknowledge the immense of the participants for cooperation during the data collection.

## Author contributions

**Conceptualization:** Benjamin Adjei, Joseph Kwame Mintah, Thomas Hormenu.

**Data curation:** Benjamin Adjei, Joseph Kwame Mintah, Thomas Hormenu.

**Formal analysis:** Benjamin Adjei, Joseph Kwame Mintah, Thomas Hormenu.

**Funding acquisition:** Benjamin Adjei, Joseph Kwame Mintah.

**Investigation:** Benjamin Adjei, Thomas Hormenu.

**Methodology:** Benjamin Adjei, Joseph Kwame Mintah, Thomas Hormenu.

**Project administration:** Benjamin Adjei.

**Resources:** Benjamin Adjei.

**Software:** Benjamin Adjei, Thomas Hormenu.

**Supervision:** Joseph Kwame Mintah, Thomas Hormenu.

**Validation:** Joseph Kwame Mintah.

**Visualization:** Benjamin Adjei.

**Writing – original draft:** Benjamin Adjei, Joseph Kwame Mintah.

**Writing – review & editing:** Benjamin Adjei, Joseph Kwame Mintah, Thomas Hormenu.

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
