## [Decision Letter · Decision Letter 0]

25 Jul 2024

PGPH-D-24-01347

Community-based Surveillance System Structures and Knowledge Level of Community-based Surveillance Volunteers on Priority Diseases in Ghana

Dear Dr. Hormenu,

Thank you for submitting your manuscript to PLOS Global Public Health. After careful consideration, we feel that it has merit but does not fully meet PLOS Global Public Health’s publication criteria as it currently stands. Therefore, we invite you to submit a revised version of the manuscript that addresses the points raised during the review process.

The reviewers have suggested major revision. We invite the authors to address the concerns raised on the manuscript.

We look forward to receiving your revised manuscript.

Kind regards,

Sunday Adedini, PhD

Academic Editor

Journal Requirements:

Additional Editor Comments (if provided):

Reviewers' comments:

Reviewer's Responses to Questions

**Comments to the Author**

1. Does this manuscript meet PLOS Global Public Health’s publication criteria ? Is the manuscript technically sound, and do the data support the conclusions? The manuscript must describe methodologically and ethically rigorous research with conclusions that are appropriately drawn based on the data presented.

Reviewer #1: Partly

Reviewer #2: Partly

2. Has the statistical analysis been performed appropriately and rigorously?

Reviewer #1: Yes

Reviewer #2: Yes

3. Have the authors made all data underlying the findings in their manuscript fully available (please refer to the Data Availability Statement at the start of the manuscript PDF file)?

Reviewer #1: Yes

Reviewer #2: No

4. Is the manuscript presented in an intelligible fashion and written in standard English?

Reviewer #1: Yes

Reviewer #2: Yes

5. Review Comments to the Author

Reviewer #1: Abstract:

Methodology:

The abstract mentions a “cross-sectional descriptive survey” but doesn’t specify how the knowledge assessment of CBSVs was conducted. Were they given a test, interviewed, or assessed through some other method?

Results:

The abstract states that CBSVs have “low knowledge” about priority diseases, but it doesn’t provide concrete data or examples to support this claim. What percentage of CBSVs couldn’t identify neonatal tetanus or CSM? What were the specific knowledge gaps?

Conclusion:

The abstract mentions that CBSVs who received resources or had committees had better knowledge. This suggests a potential bias in the results. Were these factors controlled for in the analysis? If not, the conclusion that the system is “weak” might be an oversimplification.

The abstract’s conclusion calls for general measures like “stakeholder collaboration” and “capacity building.” Based on the study’s findings, what specific training or resources should be provided to CBSVs? The conclusion should provide actionable recommendations.

Introduction:

The introduction lack focus. First, it discusses various aspects of CBS, such as its importance, the role of volunteers, and the WHO’s evaluation indicators. However, it does not clearly identify the specific problem that the research will address.

The introduction should first give a global overview and regional overview of CBS, summarise the key concepts of CBS, focusing on aspects most relevant to the study which is on knowledge level of community-based surveillance volunteers. Also, define key terms and concepts when they are first introduced to ensure the reader understand their meaning.

The study needs to clearly highlight the limitations of previous studies and unanswered questions they left behind to show the need for current research. The introduction mentions that “factors contributing to these are not known and documented” but does not specify what these factors might be. Provide potential factors that might contribute to the lack of active surveillance.

The discussion of the specific issue in the Central Region of Ghana should be moved to the middle of the introduction. This sets the context for the study and helps the reader understand its purpose.

The last sentence mentions the purpose of the study, but it is too broad. It mentions evaluating CBS structures, assessing knowledge levels, and determining differences in knowledge. These are three separate objectives, and it’s not clear how they are connected, or which is the primary focus.

Methodology:

Study design:

Provide a more detailed rationale for why a cross-sectional descriptive study design is suitable for answering the research questions and the specific advantages it offers for this study.

Study area:

The description of the study area lacks context. Explain why the Central Region of Ghana was selected for this study and how its characteristics make it a relevant area for the research.

Sampling:

The sampling strategy and sample size calculation are described but lack clarity in how the specific sample size of 1,381 CBSVs was determined. Provide a clear formula or detailed explanation for the sample size calculation, including any statistical considerations such as confidence levels, margins of error, and population proportions.

Data collection techniques:

Break down the data collection process into clear steps and describe each step methodically. Include details on the questionnaire design, pilot testing (if any), and how data quality was ensured during collection.

The questionnaire asks CBSVs to recall case definitions from their registers. This could introduce recall bias, as CBSVs might not accurately remember all the details or might be influenced by recent events or training.

Provide more details on the training content, duration, and specific topics covered. Explain how the training ensured the reliability and validity of the data collected.

Data analysis

The use of chi-square tests and logistic regression is appropriate for examining associations, the paper does not provide information on how the assumptions of these tests were checked. For instance, were the logistic regression models checked for multicollinearity? While chi-square tests and logistic regression are appropriate, the analysis could be more comprehensive. Exploring potential interactions between variables, such as employing multilevel modelling techniques could account for the hierarchical structure of the data (district, community, CBSV).

Scoring interpretation

The scoring system for assessing the knowledge level of CBSVs is described but is confusing. Use a table or diagram to visually represent the scoring system and interpretation criteria. This will make it easier for readers to understand the scoring process.

Ethics:

More details regarding the informed consent process should be given. Explain how participants were informed about the study, how consent was obtained, and any measures taken to ensure confidentiality and data protection.

Results:

Table 5

Collapsing the knowledge level variable (originally with four categories: poor, average, good, excellent) into a binary variable (weak vs. better) results in loss of information and potentially reduces the statistical power of the analysis. A more appropriate approach would be to use ordinal logistic regression, which can model the ordinal nature of the outcome variable.

Can you include the intext citation of table 6.

Discussion:

While the discussion references some previous studies, it lacks a comprehensive and critical integration of relevant literature. The authors should more thoroughly compare and contrast their findings with existing research to identify areas of agreement, disagreement, and unique contributions.

Why does receiving resources and having committee meetings lead to better knowledge among CBSVs? What are the specific challenges faced by CBSVs in communities with weak surveillance structures? A more in-depth analysis would enhance the understanding of the issue.

The claim that weak structures affect health monitoring and response needs to be substantiated with concrete examples from the study findings or relevant literature.

The recommendations provided in the conclusion are general and lack specificity. Instead of broad suggestions like “stakeholder collaboration” and “capacity building,” the authors should propose concrete, actionable recommendations based on the study’s specific findings. For example, what type of training should be provided to CBSVs? How can resource allocation be improved?

Recommendations

Stating that stakeholders should “continuously review and evaluate the CBS system structures” does not provide guidance on how this should be done, who should be involved, or what specific aspects of the system need improvement.

The recommendations focus primarily on the roles of health authorities and institutions, neglecting the importance of community participation and ownership in strengthening CBS. The community’s perspectives and needs should be central to any recommendations for improvement.

Reviewer #2: Summary:

Introduction could be improved by proceeding with standard CBS system (global standard). The components features of standard CBS system should be highlighted, then followed with local context’s CBS system operation requirements. Introduction is deficient on priority diseases highlights and the need for its knowledge among CBSVs. Also, establishing link between real or potential consequences of inadequate priority diseases knowledge among CBSVs and CBS system would also set the stage for explaining the study findings. Authors appeared to ignore other components of CBS system structure which include organization framework, data documentation tools, data management, channels of communication and response plans among others. Rather, there appeared to be an exclusive focus on the study outcome – knowledge.

Overall, while the issue of focus is topical, there needs to be an extensive incorporation of issues on the first variable of the topic - Community-based Surveillance System Structures - and improve linkage with knowledge levels.

Page 1

Para1, Line4: Response recommends that volunteers… (No earlier mention of voluneers before this statement and therefore the “the” may not be relevant

Para1, Line 10: Studies have shown… (These studies were not referenced)

Page 2

Para 2: It would have a better relevance if the focus of the study – CBS system structure and CBSVs’ knowledge is situated in the study area

Para 2, Line 4: First mention of CHPS was not explained

Para 4, Line 2: First mention of DHMT was not explained

Para 5, Line 1: A structured questionnaire …

Page 3

Para 1: First mention of CHO was not explained

Para 3, Line 8: As the practice in scientific writing, I would expect the situation of study objectives towards the end of the introduction

Para 6, Line 1: Chi square tests are appropriate for either “independence” or “association”, they are not used for “difference” tests.

Para 6, Line 3: Odds Ratios are associative effects tests, used for examining ratio of odds of an event occurring against another i.e. interpretation similar to likelihoods. With p-values and confidence intervals, significance of association can be determined, however, strength of such association are not determined by these estimates.

Page 5

Table 1: Frequencies and percentages of respondents’ occupation do not align with the report

Page 9

Table 5: While the table and its title aptly tested the “association” between supportive surveillance function and knowledge, the title of its report spoke to “difference”. This should be corrected to align with the relevant results.

Footnote of Table 5 included Fishers’ Exact test as an alternative to Chi-square test. However, since none of the table cells had entry with frequency less than 5, Fisher’s test may not be relevant here.

6. PLOS authors have the option to publish the peer review history of their article (what does this mean? ). If published, this will include your full peer review and any attached files.

**Do you want your identity to be public for this peer review?** For information about this choice, including consent withdrawal, please see our Privacy Policy .

Reviewer #1: No

Reviewer #2: No

---

## [Editor Report · Decision Letter 1]

23 Sep 2024

PGPH-D-24-01347R1

Community-based Surveillance System Structures and Knowledge Level of Community-based Surveillance Volunteers on Priority Diseases in Ghana

Dear Dr. Hormenu,

Thank you for submitting your manuscript to PLOS Global Public Health. After careful consideration, we feel that it has merit but does not fully meet PLOS Global Public Health’s publication criteria as it currently stands. Therefore, we invite you to submit a revised version of the manuscript that addresses the points raised during the review process.

We invite the authors to address the reviewer comments and resubmit the revised manuscript.

Please ensure that your decision is justified on PLOS Global Public Health’s publication criteria  and not, for example, on novelty or perceived impact.

We look forward to receiving your revised manuscript.

Kind regards,

Sunday Adedini, PhD

Academic Editor
---

## [Editor Report · Decision Letter 2]

3 Dec 2024

PGPH-D-24-01347R2

Community-based Surveillance System Structures and Knowledge Level of Community-based Surveillance Volunteers on Priority Diseases in Ghana

Dear Dr. Hormenu,

Thank you for submitting your manuscript to PLOS Global Public Health. After careful consideration, we feel that it has merit but does not fully meet PLOS Global Public Health’s publication criteria as it currently stands. Therefore, we invite you to submit a revised version of the manuscript that addresses the points raised during the review process. Please see below some additional editor comments to ensure that the manuscript meets our publication criteria.  

We look forward to receiving your revised manuscript.

Kind regards,

Ruth Ashton, Ph.D.

Academic Editor

Journal Requirements:

Additional Editor Comments (if provided):

Thank you for the revisions to respond to the peer reviewers. I have some additional modifications to request to ensure alignment with PLoS Global Public Health structure and maximise clarity of your methods.

1. Please revise the abstract to 300 words length (or as close as possible). The abstract should mention techniques used without going into methodological detail and should summarize the most important results. While I understand that some edits were made to the abstract to include suggestions from reviewers, particularly adding the recommendations section, unfortunately it is necessary to reduce the abstract length. Please also note that subheadings are not used in the abstract, although conceptually the abstract is divided into three sections (Background, Methods/Principal findings, and Conclusions/Significance).

2. The introduction should be proof-read to correct some sense and typing errors, and to ensure that information is concisely presented. Some extraneous statements that are not directly related to the study (such as the citation on knowledge of meningitis by travelers from the UK to the meningitis belt) could also be removed.

3. In the methods, please clarify what is meant by a systematic random sample, since these terms appear to be contradictory – sampling is usually either systematic or random.

4. In the methods, data analysis section, please add a sentence clarifying how you reclassified your level of knowledge from 4 classes to a binary outcome for the logistic regression.
---

## [Editor Report · Decision Letter 3]

6 Jan 2025

Community-based Surveillance System Structures and Knowledge Level of Community-based Surveillance Volunteers on Priority Diseases in Ghana

PGPH-D-24-01347R3

Dear Dr Hormenu,

We are pleased to inform you that your manuscript 'Community-based Surveillance System Structures and Knowledge Level of Community-based Surveillance Volunteers on Priority Diseases in Ghana' has been provisionally accepted for publication in PLOS Global Public Health.

Best regards,

Ruth Ashton, Ph.D.

Academic Editor